# Decoding IL-23 Signaling Cascade for New Therapeutic Opportunities

**DOI:** 10.3390/cells9092044

**Published:** 2020-09-07

**Authors:** Gloria Pastor-Fernández, Isabel R. Mariblanca, María N. Navarro

**Affiliations:** Centro de Biología Molecular Severo Ochoa, Consejo Superior de Investigaciones Científicas and Universidad Autónoma de Madrid (CSIC/UAM), 28049 Madrid, Spain; gloria.pastor@cbm.csic.es (G.P.-F.); isabel.rguez.mb22@gmail.com (I.R.M.)

**Keywords:** interleukin 23, IL-23, signaling, interleukin 17, autoimmunity, inflammatory disease, psoriasis, multiple sclerosis

## Abstract

The interleukin 23 (IL-23) is a key pro-inflammatory cytokine in the development of chronic inflammatory diseases, such as psoriasis, inflammatory bowel diseases, multiple sclerosis, or rheumatoid arthritis. The pathological consequences of excessive IL-23 signaling have been linked to its ability to promote the production of inflammatory mediators, such as IL-17, IL-22, granulocyte-macrophage colony-stimulating (GM-CSF), or the tumor necrosis factor (TNFα) by target populations, mainly Th17 and IL-17-secreting TCRγδ cells (Tγδ17). Due to their pivotal role in inflammatory diseases, IL-23 and its downstream effector molecules have emerged as attractive therapeutic targets, leading to the development of neutralizing antibodies against IL-23 and IL-17 that have shown efficacy in different inflammatory diseases. Despite the success of monoclonal antibodies, there are patients that show no response or partial response to these treatments. Thus, effective therapies for inflammatory diseases may require the combination of multiple immune-modulatory drugs to prevent disease progression and to improve quality of life. Alternative strategies aimed at inhibiting intracellular signaling cascades using small molecule inhibitors or interfering peptides have not been fully exploited in the context of IL-23-mediated diseases. In this review, we discuss the current knowledge about proximal signaling events triggered by IL-23 upon binding to its membrane receptor to bring to the spotlight new opportunities for therapeutic intervention in IL-23-mediated pathologies.

## 1. Introduction

The interleukin 23 (IL-23) has emerged as a key pro-inflammatory cytokine involved in the development of chronic inflammatory diseases, such as psoriasis, inflammatory bowel diseases, multiple sclerosis, or rheumatoid arthritis in murine models [1] and, more importantly, in humans. For example, increased levels of IL-23 have been found in biopsies from patients with Crohn’s disease, ulcerative colitis, and psoriasis. Moreover, the interference with IL-23 signaling pathway using neutralizing antibodies has shown efficacy in different inflammatory conditions [1,2].

IL-23 is a pro-inflammatory cytokine that belongs to the IL-12 family, together with IL-12, IL-27, IL-35, and IL-39 [3]. IL-23 is produced by dendritic cells and activated macrophages [4], and its actions are mainly mediated by the CD4 T helper subset Th17, a distinct subpopulation of gamma/delta T cells (Tγδ17 cells), subsets of natural killer T (NKT) cells, and type 3 innate lymphoid cells (ILC3s) [1] (Figure 1). Chronic inflammatory diseases are initiated by uncontrolled immune responses that trigger the hyperactivation of signaling cascades, promoting the accumulation of pro-inflammatory mediators and, finally, the manifestation of clinical symptoms. The pathological consequences of excessive IL-23 signaling have been linked to its ability to promote the production of inflammatory mediators, such as IL-17, IL-22, the granulocyte-macrophage colony-stimulating factor (GM-CSF), or the tumor necrosis factor (TNFα) by target populations [5,6,7,8,9]. These pathogenic mediators promote the rapid recruitment and activation of granulocytes and macrophages, equipped with an extensive tool set to cause the tissue damage that induce chronic inflammation and, finally, the development of clinical symptoms. To date, several reports have shown an important role for IL-23 and its pathogenic mediators in murine models of rheumatoid arthritis, psoriasis, spondyloarthropathy, and inflammatory bowel diseases [10] (Figure 1).

IL-17 was the first effector molecule identified downstream of IL-23 [5]. Of the six members of the mammalian IL-17 family (IL-17A-IL-17F), IL-17A and IL-17F have been linked to chronic inflammatory diseases. IL-17A and IL-17F bind as disulfide-linked homo or heterodimers to a receptor complex formed by the IL-17RA and IL-17RC subunits [11]. IL-17RA is expressed on many non-immune cells, such as endothelial cells, epithelial cells, and fibroblasts, and by immune cells including T cells [2]. Upon IL-17 binding, the adaptor protein Act1 is recruited to the receptor’s cytoplasmic tail and the E3 ligase activity of Act1 ubiquitylates the tumor necrosis factor receptor-associated factor 6 (TRAF6) to initiate the activation of the canonical nuclear factor kappa-light-chain-enhancer of activated B cells (NF-κB), mitogen-activated protein kinase (MAPK) pathways, and the CCAAT/enhancer-binding protein (C/EBP) family of transcription factors [11]. IL-17 activates epithelial cells, fibroblast, and keratinocytes and increases the secretion of antimicrobial peptides, cytokines, and chemokines that further recruit and activate mast cells, neutrophils, and inflammatory macrophages [12,13]. Chemokines induced by IL-17 promote the recruitment of neutrophils, Th17 cells, and ILC3s [14], and IL-17-mediated production of IL-6 and G-CSF further activates neutrophils and other myeloid cells [15]. In rheumatoid arthritis, IL-17 action on different joint cells induced the production of TNF, IL-1β, and GM-CSF to promote bone and cartilage degradation [16]. IL-17-responsive skin-resident cells also produce autocrine factors that promote cell division and neo-angiogenesis. Additionally, IL-17 is a strong inducer of antimicrobial peptides, such as β-defensins and Lipocalin 2, which can prevent infection and bacterial growth at mucosal surfaces and in the skin [17,18]. Thus, the interaction between IL-17-secreting cells and tissue-resident keratinocytes, epithelial cells, and fibroblasts creates amplification loops that drive local inflammation, tissue damage, and finally lead to the manifestation of clinical symptoms (Figure 1). In addition to its pro-inflammatory function, IL-17 has a key role in the maintenance of barrier properties of epithelial tissues [19], and clinical trials targeting IL-17A and IL-17RA in patients with Crohn’s disease had to be terminated due to exacerbation of colitis [20,21]. This outcome is consistent with the elevated Th17 cell response in the intestinal mucosa in the steady state, which is possibly directed against commensal bacterial antigens. The protective role of IL-17 has been linked to the regulation of the microbiota [22], to the maintenance of intestinal epithelial cell tight junctions and barrier integrity [23,24], and to the stimulation of repair upon intestinal epithelial damage [25]. IL-22 is another relevant IL-23 downstream effector molecule [6]. IL-22 belongs to the IL-10 family of cytokines, and IL-22 active monomers bind to the IL-22 receptor complex (IL-22R) formed by the IL-22R1 and IL-10R2 subunits. IL-22 binding to the IL-22R promotes the activation of the Janus tyrosine kinases (JAKs) Jak1/Tyk2, leading to the signal transducer and activator of transcription 3 (STAT3) phosphorylation. IL-22 binding also activates the MAPK and p38 pathways [26]. IL-22 is expressed in a broad array of tissues, and its receptor is expressed on the stromal and epithelial cells of those tissues. IL-22 promotes the regeneration of epithelial tissues upon injury mainly inducing proliferation and inhibiting apoptosis of epithelial cells, and this same function has been implicated in pathological states. In addition, a key role for IL-22 has been described in host defense within barrier tissues, such as the intestine, oral mucosa, skin, and lung [27,28]. IL-22 has pleiotropic effects on keratinocytes, such as induction of proliferation, migration, tissue remodeling, and secretion of antimicrobial peptides, cytokines, and chemokines, as well as delayed differentiation [29]. IL-22 promotes the production of neutrophil-attracting chemokines [30] and the expression of extracellular matrix metalloproteinases, which are required for tissue remodeling during epithelial repair [31]. Thus, IL-22 stimulation of epithelial cell survival and proliferation during acute tissue damage can be protective, while excessive production of IL-22 could lead to keratinocyte hyperproliferation, production of pro-inflammatory signals, and subsequent recruitment of pathologic effector cells to the inflamed tissues (Figure 1). Intestinal IL-22 is critical for immunity against *Citrobacter rododentium* [32,33], and it induces expression of genes regulating proliferation, wound healing, and apoptosis of intestinal epithelial cells [34]. In addition to its role in host defense, IL-22 provides functional barrier support through induction of cell proliferation, mucins, and antimicrobial peptides [35]. In fact, the interference with the IL-22/IL-22R pathway exacerbated colitis in some mouse models [36,37]. Thus, as for IL-17, both pro-inflammatory and tissue-protective functions have been identified for IL-22. Interestingly, the role in intestinal homeostasis of Th17-derived IL-17 and IL-22 are independent of IL-23 [23,24,38], and thus, the development of selective IL-23 inhibitors hold the promise to interfere especially with pathogenic IL-17-producing cells without affecting maintenance of the gut barrier. GM-CSF has emerged as the key pathogenic effector molecule downstream of IL-23 in the development of the experimental autoimmune encephalomyelitis (EAE) model of multiple sclerosis [7,8]. GM-CSF is secreted as a monomeric cytokine that binds to the GM-CSF receptor, a heterodimer formed by a specific α subunit and a common beta (βc) subunit shared with IL-3 and IL-5 receptors. GM-CSF binding to its cognate receptor promotes the activation of Jak2 and subsequent STAT5 phosphorylation, Src family kinases, and the phosphatidylinositol 3-kinase (PI3K) and mitogen-activated protein kinase (MAPK) pathways. The main GM-CSF responder populations are dendritic cells, monocytes, macrophages, granulocytes, neutrophils, and importantly, microglia and astrocytes [39,40]. Despite its initial classification as a hematopoietic growth factor, GM-CSF plays a minor role in myelopoiesis, and it is emerging as a major mediator of tissue inflammation. GM-CSF induces a genetic program involved in inflammasome function, phagocytosis and chemotaxis that participate in tissue destruction and demyelination [41]. GM-CSF promotes monocyte migration from the bone marrow across the hematoencephalic barrier and into the central nervous system (CNS) [42]. Once at the CNS, GM-CSF promotes the differentiation of infiltrating monocytes into antigen presenting cells that contribute to the maintenance of the pathogenic Th17 cells [43] and also induces production of pro-inflammatory mediators that promote tissue damage, demyelination, and axonal loss [44]. Finally, although less studied than IL-17, IL-22, and GM-CSF, IL-23 also induces the production of TNFα, IL-19, and IL-24 in a skin inflammation model [9].

IL-23 is required to provide effective host defense against a wide variety of extracellular pathogens, such as bacteria, parasites, fungi, and viruses [1]. However, due to their pivotal role in inflammatory diseases, IL-23 and its downstream effector molecules have emerged as attractive therapeutic targets. The emergence of neutralizing antibodies against harmful pro-inflammatory mediators has marked a milestone in the development of new therapeutic strategies. In this context, blocking antibodies against IL-23 and IL-17 have been approved for treatment of plaque psoriasis, and they are currently under Phase II/Phase III clinical trials for inflammatory bowel diseases, multiple sclerosis, and rheumatoid arthritis [1]. Therapeutic interventions using blocking antibodies in the context of IL-23-mediated diseases have been recently and extensively reviewed elsewhere [2,11,45,46,47]. Despite the success of monoclonal antibodies, not all patients respond to these treatments, and others show a partial response. Thus, effective therapies for chronic inflammatory diseases may require the combination of multiple immune-modulatory drugs to prevent disease progression and to improve quality of life. Alternative strategies aimed at inhibiting intracellular signaling cascades using small molecule inhibitors or interfering peptides have not been fully exploited in the context of IL-23-mediated diseases. The interference with intracellular signaling cascades has been successfully applied for the treatment of different types of cancer and inflammatory pathologies [48,49]. In comparison to monoclonal antibodies, small molecule inhibitors have a broader tissue distribution, possibility of development of oral/topical versions, and reduced production costs [50]. These therapies are effective, economic, and thus, suitable for mild clinical symptoms or to be used in combination with monoclonal antibodies therapies. In addition, engineered, non-immunoglobulin protein scaffolds that interfere with IL-23 or the IL-23R represent another therapeutic strategy for treatment of chronic inflammatory diseases. Protein scaffolds are based in natural proteins and use combinatorial protein engineering to change their affinity and specificity to bind and block a desired molecule. This process results in the generation of small, stable, single-chain proteins with high-affinity binding sites [51]. These protein scaffolds are often produced in *Escherichia coli*, being less costly and time-consuming to process than monoclonal antibody production with no post-transcriptional modifications, and its reduced size presents an advantage in terms of tissue distribution. Hence, these scaffolds offer structural plasticity to create novel molecules with specific binding properties designed against relevant effector molecules. Here, we have focused on the proximal signaling cascade triggered by IL-23 upon binding to its membrane receptor to bring to the spotlight new opportunities for therapeutic intervention in IL-23-mediated pathologies.

## 2. IL-23/IL-23R Structure and Binding

IL-23 is a heterodimeric cytokine formed by two disulfide-linked bound subunits, p40 (encoded by *Il12b*) and p19 (*Il23a*) [4], that bind to a membrane receptor complex formed by two chains: the IL-12Rβ1 (*Il12rb1*) and the unique IL-23R (*Il23ra*). IL-23 shares the p40 subunit and one of its receptor components, the IL-12Rβ1, with the IL-12 [52]. Despite the shared subunits, IL-23 and IL-12 have distinct and unique biological functions [53].

The crystal structure of IL-23 shows that the p40 subunit is composed of three domains (D1, D2 and D3) [54], while p19 is a four-helix bundle cytokine [55]. The disulfide bond is formed between p19-Cys54 and the p40-Cys177 residues, and the binding is primarily mediated by the p19-Arg159 residue that forms an extensive network of interactions with different residues on p40 [55] (Protein Data Bank IDs for crystal structure of IL-23: 3DUH, 5MXA, 3D87). The mutation of p19-Arg179 (Arg159 in [55]) or key Tyr residues in the p40 subunit abolished the formation of biological active IL-23 [56]. The secretion of the biologically active form of IL-23 requires the co-expression of p19 and p40 subunits, encoded on different chromosomes, within a cell [4]. Mechanistically, it has been recently shown that partially unfolded p19 subunit is recognized and retained by chaperones at the endoplasmic reticulum. Upon binding to the p40 subunit, p19 completes its folding, inhibiting chaperone interaction and resulting in the secretion of the heterodimeric IL-23 [57]. IL-23 p19 subunit also associates with the Epstein-Barr virus-induced gene3 (EBi3) subunit to form the recently discovered IL-39 implicated in experimental murine lupus in mice [58], although IL-39 expression in humans has yet to be detected [59].

IL-23 binds to a membrane receptor complex formed by two type I membrane proteins: the IL-12Rβ1 and the IL-23R. IL-12Rβ1 contains two extracellular cytokine receptor domains and three fibronectin-type III domains, followed by a single transmembrane domain and a cytoplasmic domain [60,61]. IL-23R contains an N-terminal immunoglobulin-like domain, two cytokine receptor domains, a single transmembrane domain, and a cytoplasmic domain [3,52]. The p40 subunit binds to the IL-12Rβ1 and p19 to the IL-23R chain, inducing receptor oligomerization and promoting the formation of a quaternary signaling complex in a 1:1:1:1 stoichiometry [4,52,56]. Recently, the crystal structure of human IL-23 bound to the IL-23R has shown that the p19 subunit binds to IL-23R via the p19 N-terminal immunoglobulin domain, restraining the p40 subunit to recruit the IL-12Rβ1 to the signaling complex [62] (Protein Data Bank ID for crystal structure of IL-23/IL-23R: 5MZV). The p19 subunit contains a conserved hot spot site III Tryptophan at position 156 (Trp157 in the mouse p19 subunit), and point mutations of p19-Trp-156 disrupted the interaction with IL-23R. Accordingly, a mIL-23p19-Trp157Ala mutant was unable to drive the production of IL-17a and IL-22 in Th17 cells [62]. Moreover, while the subcutaneous injection of mIL-23p19WT induced the thickening of epidermis and leukocyte infiltration [5,6,7,8,9], administration of mIL-23p19Trp157Ala mutant exhibited a complete lack of pathologies [62]. Thus, the interaction of both p40/IL-12Rβ1 and p19/IL-23R is mandatory for the formation of a signaling competent IL-23R complex [56,62]. The accumulated knowledge about IL-23/IL-23R structure licenses the engineering of biological and pharmacological compounds with antagonistic behavior against IL-23/IL-23R and, thus, with potential to interfere with the intracellular signaling cascade that leads to the production of harmful mediators. For example, an IL-23R peptide antagonist was shown to reduce inflammation parameters in mouse models of skin inflammation induced by subcutaneous injection of IL-23, in systemic inflammation induced by anti-CD40 and in collagen-induced arthritis (CIA) [63] (Table 1). Additionally, protein scaffolds called Alphabodies have been engineered to interfere with the p-19-Trp156/157 residue that is key for the interaction between p19 and the IL-23R. This protein scaffold blocked IL-23 signaling in vitro and more importantly, in vivo, in a model of skin inflammation induced by intradermal injection of recombinant IL-23 [64] (Table 1). Engineered combinatorial libraries derived from the albumin-binding domain (ABD) of streptococcal protein G30–33 were used to generate recombinant proteins that bind the hIL-23R (REX binders) [65] and the p19 subunit of hIL-23 (ILP binders) [66]. Both REX and ILP binders were able to compete with p19 binding to IL-23R in vitro, and inhibited IL-23-mediated expansion of primary human Th17 cells [65,66]. This ABD-based strategy has been also used to generate IL-17a binders (ARS binders) that prevented hIL-17A binding to its cell-surface receptor [67]. Taking this research one step closer to application in humans, recombinant strains of *Lactococcus lactis* expressing REX009 and REX115 ligands (hIL-23R binders) [68] and LP317 ligand (hIL-23 binder) [69] have been created. These IL-23R/IL-23 binders were secreted and retained the ability to bind IL-23R or IL-23 [68,69]. Thus, although these strategies require further testing in disease models of IL-23-mediated pathologies, oral administration of these strains as a probiotic represent a promising method of delivery of the IL-23 antagonists for pathologies, such as inflammatory bowel diseases. Other approaches using hydrogen–deuterium exchange mass spectrometry (HDX-MS) identified macrocyclic small molecule against IL-23R that interfered with binding of IL-23 to the IL-23R [70]. The efficacy of this small molecule still needs to be tested in in vivo models of IL-23-mediated pathologies, but overall, the strategies aimed to interfere with IL-23/IL-23R binding using small molecules and antagonistic peptides open new gateways for specific treatment of inflammatory diseases.

## 3. IL-23 Proximal Signaling Cascade: JAKs/STATs Module

The IL-23 receptor complex, like other type I/II cytokine receptors, lacks intrinsic enzymatic activity. The IL-23 proximal signaling cascade involves the activation of members of the JAK family of tyrosine kinases, and their downstream effectors, the STAT family of transcription factors. Cytokine binding promotes the oligomerization of membrane receptors, bringing receptor-bound JAKs to a close proximity and promoting conformational changes that distance their kinase domains from inhibitory pseudokinase domains. Subsequently, JAKs activation promotes the phosphorylation of Tyr residues in both the JAKs themselves and the cytoplasmic tails of the receptors, creating docking sites for STAT monomers through their Src homology 2 domains (SH2). Active JAKs then phosphorylate STAT monomers, leading to dimerization, nuclear translocation, and DNA binding to target genes promoters [71,72] (Figure 2). IL-23 receptor complex is associated with the JAK family members Jak2 and Tyk2, whose activation predominantly promoted STAT3 phosphorylation and to a lesser extent STAT1, STAT4, and STAT5 phosphorylation [4,52]. Deletion and site-directed mutagenesis approaches identified Box1 and Box2 motifs in the intracellular domain of IL-12Rβ1 as the Tyk2 docking sites. In contrast, IL-23R lacks Box1 and Box2 motifs, and an atypical Jak2-binding site in the cytoplasmic tail of IL-23R has been defined [52,73]. Analysis of kinase-deficient cell lines showed that both Jak2 and Tyk2 are required for IL-23 signal transduction [73]. The role of Jak2 in IL-23 signaling in vivo cannot be addressed in knockout animal, as Jak2 deficiency led to embryonic lethality due to defects in erythropoiesis [74]. In contrast, Tyk2 deficient animals are viable, and in vitro differentiated Th17 from Tyk2 null mice failed to respond to IL-23. Moreover, Tyk2 deficient animals were protected against disease development in the experimental autoimmune encephalomyelitis (EAE) model of multiple sclerosis [75] and in the Imiquimod (IMQ)-induced skin inflammation [76]. Interestingly, genome-wide association studies (GWAS) identified a Tyk2 protective polymorphism in autoimmune diseases. The development of a knock-in murine model of this protective variant (Tyk2-Pro1104Ala) showed defective IL-23 signaling and protection against EAE development in the knock-in animals [77]. Collectively, these data confirm the key role of Tyk2 in the IL-23 signaling cascade.

The inhibition of JAKs catalytic activity using small molecule inhibitors represents an outstanding opportunity for therapeutic intervention. Different JAK inhibitors (Jakinibs) have been approved for treatment of lympho-proliferative malignancies, and they are already approved or in different phases of clinical trials for treatment of autoimmune and inflammatory diseases (excellently reviewed in [78,79]). A caveat of JAKs inhibition is that the catalytic domain of the different family members shares a high degree of homology. Thus, first-generation Jakinibs blocked multiple JAKs, and these pharmacological inhibitors had unwanted side effects like increased susceptibility to infection, anemia, and other hematological defects. The development of more selective next-generation Jakinibs is a very active research field. Tofacitinib, a Jak3/Jak1 inhibitor that also targets Jak2 to a lesser extent, has been approved for treatment of ulcerative colitis and psoriatic arthritis. Ruxolitinib and Baricitinib—Jak1/2 inhibitors—are currently in clinical trials for atopic dermatitis [78] (Table 1). In the context of IL-23, Tyk2 inhibition has a potential lower risk of side effects, as Tyk2 is used by less cytokine receptors than Jak2. BMS-986165 is a selective Tyk2 inhibitor that have shown efficacy in a Phase II clinical trial for psoriasis treatment [80] (Table 1), and it is being studied in systemic lupus erythematosus, psoriatic arthritis, and inflammatory bowel diseases. Brepocitinib (PF-06700841) is a Tyk2/Jak1 inhibitor that is being tested in psoriasis [81] (Table 1). Interestingly, the efficacy of topical treatments with Jakinibs is being examined in clinical trials for treatment of dermatological disorders [82]. Topical treatments can potentially reduce the risk of side effects avoiding the systemic action of the drugs, and thus, these therapies are of particular interest for skin conditions, and they can have a great impact on patient quality of life.

Upon IL-23 binding and Jak2/Tyk2 transactivation, the next step in the IL-23 signaling cascade is the recruitment of STAT3 to the receptor complex to allow phosphorylation and activation by JAKs [4,52]. STATs recruitment is mediated by the binding of STAT-SH2 domains to phosphorylated Tyr residues in the membrane receptors (Figure 2). Potential phosphotyrosine-binding sites for STATs have been found in the intracellular domain of IL-23R but not in IL-12Rβ1. The cytoplasmic domain of IL-23R contains seven tyrosine residues, and six of them are conserved between murine and human IL-23R. Experiments using single mutant variants of IL-23R-Tyr residues indicated that IL-23-mediated STAT3 recruitment and phosphorylation did not rely on one single tyrosine motif. This work found both canonical STAT3 binding sites and atypical SH2-binding sites in the IL-23R are required for STAT3 recruitment and phosphorylation [83].

The inhibition of STAT3 can be achieved by blockage of phosphorylation using Jakinibs, or through the interference with the SH2 domains. STAT3-SH2 domains are required for recruitment to cytokine receptors and for dimerization, a prerequisite for translocation to the nucleus and DNA binding (Figure 2). Although the interference with STAT3-SH2 domains has not been examined in the context of inflammatory pathologies, small molecule inhibitors that bind to STAT3-SH2 and block STAT3 phosphorylation have shown promising results in multiple myeloma and other hematopoietic malignancies [84]. Additionally, salicylic acid analogs and other small molecules have been described as potent STAT3 inhibitors with antitumor activity [85,86]. Thus, pharmacological inhibition of STAT3-SH2 stands out as a relevant therapeutic intervention to be explored in IL-23-related diseases. However, targeting STAT3 in a systemic manner is challenging, since STAT3 plays key cell and tissue specific roles both in steady state and in disease. The genetic ablation of STAT3 in all cells is embryonically lethal [87], and cell-specific STAT3 knockouts revealed key functions in myeloid and lymphoid cell differentiation and activation, liver regeneration, heart muscle development and function, development of neuronal cells, musculoskeletal system, and mammary glands [88]. The hyper-IgE syndrome (AD-HIES; also known as Job’s syndrome) is an autosomal dominant immune deficiency that is usually caused by loss-of-function mutations in STAT3 [89,90]. These patients are susceptible to recurrent pulmonary infections and chronic mucocutaneous candidiasis, *Staphylococcus aureus*, and pulmonary infections, due to a reduced frequency of Th17 cells and subsequent loss of host defense against extracellular bacteria and fungi [90,91]. These patients present aberrant B cell function, likely due to the key role of STAT3 in follicular helper T-cell differentiation [92] and interleukin-21 signaling during B-cell differentiation and activation [93]. Two possibilities for STAT3 therapeutic targeting without generating severe side effects are the partial STAT3 inhibition for a limited duration using drug-based therapies, and the delivery of the STAT3 antagonist to the desired tissues, instead of complete or systemic ablation of STAT3 function.

In addition to docking sites for STAT3-SH2 binding, Tyr residues in cytokine receptors also facilitate the attachment of the natural negative regulators of cytokine signaling: the suppressors of cytokine signaling (SOCS) proteins. SOCS-SH2 binding to cytokine receptors competes with STAT recruitment, and some SOCS members are additionally able to serve as pseudosubstrates for JAKs, thus inhibiting JAK/STAT downstream signaling cascade [3] (Figure 2). IL-23 induced the expression of Socs3 in Th17 cells [94], and Socs3-deficient Th17 cells displayed increased STAT3 phosphorylation upon IL-23 stimulation and increased IL-17 production [95]. Moreover, Socs3 expression is lower in T cells obtained from patients with psoriasis [96] and in multiple sclerosis during relapse [97] compared to healthy donors. Thus, therapeutic strategies aimed at increasing the expression of Socs3 would result in the inhibition of IL-23/JAK/STAT. For example, baicalin is a bioactive flavonoid compound obtained from *Scutellaria baicalensis*, an herb used in traditional medicine for the treatment of different inflammatory diseases, and baicalin reduced IL-23-induced phosphorylation of STAT3 and IL-17 production in Th17 cells. Baicalin treatment inhibits the capacity of Th17 to induce EAE in an adoptive transfer model, and these effects were mediated by the increase in Socs3 expression [98]. The manipulation of Socs3 expression can be also achieved using miRNAs. For example, overexpression of miR-384 that targets Socs3 resulted in exacerbated EAE and increased Th17 differentiation, while inhibition of miR-384 reversed these changes [99]. Overexpression of miR-409-3p or miR-1896 reduced Socs3 expression and increased phosphorylation of STAT3 in astrocytes in the EAE model, while the silencing of both miRNAs reduced central nervous system inflammation and demyelination [100]. Additionally, the Leukemia inhibitory factor (Lif) was shown to induce Socs3 expression in Th17 cells and to block Jak2 activation and STAT3 phosphorylation. More importantly, the inhibitory effects of Lif treatment on Th17 differentiation were lost in Socs3-silenced CD4 T cells [95]. Finally, the engineering of SOCS peptide mimetics can also interfere with JAK/STAT signaling module. Although this strategy has not been directly studied for Socs3 in the context of IL-23-signaling, Socs1 peptide mimetics that block Jak2 activation have been shown to ameliorate EAE [101] and keratinocyte inflammation [102]. Thus, the manipulation of SOCS expression or function opens a new window for therapeutic intervention in IL-23-related pathologies. In addition to STAT3, IL-23 also promotes STAT4 phosphorylation [52]. STAT4 is dispensable for the development of Th17 cells [103,104], but it contributes to maximal production of IL-17 [105,106,107] and plays a role in Th17 plasticity in late stages of EAE [108].

In addition to the triggering of proximal signaling events, IL-23 binding to its surface receptor has been recently shown to regulate IL-23R endocytosis and recycling. IL-23 stimulation of human monocyte-derived macrophages (MDMs) increased IL-23R surface expression after 15min, decreased by 30 min, increased at 1 h, and then gradually declined [109]. Using siRNAs approaches to silence protein expression, this work determined that the dynamics of IL-23R surface expression was regulated by dynamin and AP2-mediated endocytosis, followed by Rab11-dependent recycling to the cell surface. Furthermore, the interference with IL-23R recycling using siRNA against Rab11, dynamin, and AP2 reduced IL-23-dependent production of pro-inflammatory cytokines, indicating that IL-23R recycling is involved in the amplification of the IL-23 signaling cascade [109]. These studies also suggested that defective recycling contributes to the lost-of-function mechanism of the IL-23R protective polymorphism rs11209026. A key evidence of the role of IL-23 in human pathologies emerged from GWAS analysis showing that a particular coding polymorphism in the IL-23 receptor locus (rs11209026) confers strong protection against inflammatory bowel diseases [110,111] and psoriasis [112]. This variant of the IL-23R has a single amino-acid change in the intracytoplasmic domain of the IL-23R chain (R381Q), resulting in a loss-of-function allele. Using MDMs from heterozygotic carriers of the IL-23R-Q381 protective polymorphism, this variant was found to colocalize with late endosomes and lysosomal markers, while the common Il-23R-R381 variant was found in recycling compartments [109]. These results suggest that impaired recycling may be responsible for the protective action of IL-23R-Q381 polymorphism. Overall, these data indicate that the interference with IL-23R recycling represent a novel strategy for therapeutic intervention.

## 4. New Players in IL-23 Proximal Signaling Events

Recent works have shed light onto proximal events in the IL-23 signaling cascade, identifying novel players that can be targeted to manipulate cell function in inflammatory diseases. We have recently used a quantitative phosphoproteomic approach to characterize IL-23 signaling in primary murine Th17 cells. This study showed that IL-23 induced the phosphorylation of the myosin regulatory light chain Ser20 residue (RLC/MLC-Ser20) [113], an actomyosin contractibility marker [114]. RLC phosphorylation upon IL-23 stimulation required Jak2 and Rho-associated protein kinase (ROCK) catalytic activities, and the IL-23/ROCK/pRLC signaling pathway is conserved in naïve and pathogenic Tγδ17 and Th17 cells obtained from IMQ-induced skin inflammation and EAE models, respectively. Further studies revealed that IL-23 induces spontaneous migration of Tγδ17 and Th17 cells in a ROCK dependent manner, and that ROCK catalytic activity is required for recruitment of Tγδ17 to inflamed skin upon challenge with inflammatory agent IMQ [113] (Figure 3). In this context, ROCK activity has been linked to IL-23-related pathologies [115]. Particularly, administration of pharmacological inhibitors of ROCK, such as Fasudil [116,117], WAR5 compound [118], or statins [119] ameliorated disease severity and neuroinflammation in EAE (Table 1). Moreover, the administration of the oral ROCK inhibitor KD025 to psoriasis vulgaris patients reduced the severity score, epidermal thickness, and T-cell infiltration [120] (Table 1).

The molecular mechanisms that promote ROCK activation downstream of IL-23 have not been fully elucidated, but they probably involve Jak-mediated phosphorylation of guanine nucleotide exchange factors (Rho-GEFs) that trigger the activation of the Rho family of small guanosine triphosphatases (Rho-GTPases) [113]. In this context, the Dedicator of cytokinesis 8 (DOCK8), a GEF that interacts with the Rho-GTPase Cdc42, has been also linked to IL-23 signaling. IL-23-mediated STAT3 phosphorylation and IL-22 production were severely impaired in DOCK8-deficient ILCs [121] (Figure 3). As previously shown for IL-23 null mice [122], DOCK8 deficient animals were highly susceptible to *Citrobacter rodentium* infection. Lower response to IL-23 was also observed in human ILC3 obtained from DOCK8 deficient patients compared to healthy donors [123]. Mechanistically, it was shown that DOCK8 coimmunoprecipitated with STAT3 in T cells, and this association was suggested to protect STAT3 from dephosphorylation [124]. Thus, the interference with DOCK8/STAT3 binding can potentially inhibit an IL-23 signaling cascade in inflammatory pathologies.

The role of IL-23 in the regulation of cell migration may be broader than previously appreciated. A recent study in antigen-induced arthritis model (AIA) has revealed that Th17 from IL-23R deficient animals accumulate in lymphoid tissues rather than in the inflammation site [125]. T cell migration to the site of inflammation is regulated by different chemokine and chemokine receptors. In the AIA model, the expression of the C-C motif chemokine receptor type 6 (CCR6) is required for migration towards the joints, but CCR6 expression was not affected in IL-23R deficient Th17. In contrast to CCR6, CCR7 expression needs to be downregulated for activated T cell egress from lymphoid organs to inflamed site. Interestingly, IL-23R deficient Th17 cells expressed higher levels of CCR7 compared to wild-type cells and in vitro stimulation with IL-23 reduced the expression of CCR7. Thus, in addition to inducing the secretion of harmful mediators, IL-23 signaling may increase the pathogenic potential of effector cells by promoting egress from lymphoid tissues through the regulation of CCR7 expression [125] and induce cytoskeletal forces that facilitate cell migration through ROCK activation [113]. IL-23 regulation of cell migration represents a new opportunity for therapeutic interference: the blockage leucocyte migration to the inflamed site. These approaches have been proven effective for treatment of inflammatory diseases. For example, fingolimod (FTY720), a drug that induces sphingosine-1-phosphate receptor (S1PR1) down-modulation and inhibits egress of T cells from lymphoid tissues [126], is commonly used for treatment of multiple sclerosis [127]. Fingolimod has been also shown to impair Tγδ migration to the inflamed site in the IMQ model of skin inflammation [128] (Table 1). Interestingly, oral treatment with fingolimod is approved for multiple sclerosis treatment, and there are topical versions of this drug that are of interest for treatment of dermatological conditions. Natalizumab, a monoclonal antibody against alpha4 integrin (VLA4) that prevents leucocyte migration into organs, is approved for treatment of multiple sclerosis and Crohn’s disease [129] (Table 1). Further studies on how IL-23 regulates cell migration can lead to the development of treatments to specifically target migration of IL-23-responding cells.

A phosphoproteomic study of IL-23 signaling in the IL-23R-expressing human cell line Kit225 revealed that IL-23 triggered the phosphorylation of pyruvate kinase isoform M2 Ser37 residue (PKM2-Ser37), promoted its nuclear translocation, and induced the expression of PKM2 downstream target genes, such as the hypoxia inducible factor 1 subunit alpha (HIF1α) and the lactate dehydrogenase A (LDHA) [130] (Figure 3). These signaling events still need to be examined in primary cells but interestingly, pharmacological and genetic manipulation of PKM2 activity has recently shown protective effects in the EAE model [131,132]. TEPP-46 is an allosteric activator of PKM2 that induces its tetramerization, blocks its nuclear translocation, and increases its canonical enzymatic activity. The treatment with TEPP-46 reduced the expression of c-myc, HIF1α and the mechanistic target of rapamycin complex 1 (mTORC1) signaling and prevented the engagement of glycolysis in CD4 T cells, severely impairing the generation of Th17 and Th1 cells in vitro. More importantly, TEPP-46 administration delayed the development of EAE and reduced the clinical score [132] (Table 1). Although this study does not place PKM2 directly downstream of IL-23, the data suggest an interesting link between IL-23 and the regulation of cellular metabolism that deserves further study, as the metabolic reprogramming of immune cells offers a myriad of opportunities for therapeutic intervention [133].

The aryl hydrocarbon receptor (AhR) has been linked to the IL-23 signaling pathway. AhR is a ligand-dependent transcription factor, activated by pollutants (dioxin) and small molecules provided by the diet, microbiota, and cellular metabolism (i.e., tryptophan-derived metabolites, indoles, and flavonoids). Upon ligand binding, Ahr translocates into the nucleus and binds to the AhR nuclear translocator (ARNT/HIF1β) to directly promote gene transcription. AhR also binds to chromatin remodeling complexes [134,135]. AhR-deficient mice displayed a delayed onset in EAE development, and the administration of the AhR agonist FICZ increased EAE clinical score [136]. Th17 and Tγδ17 cells express high levels of AhR, and these subpopulations showed defective IL-22 production upon IL-23 stimulation in AhR null mice, suggesting that IL-23 signaling requires AhR transcriptional activity [137,138]. The expression of the activation marker CD69 represents a potential link between IL-23 and AhR activation. CD69 associates with the system L1 amino acid transporters CD98 (*Slc3a2*) and LAT1 (*Slc7a5*) that mediate amino acids uptake including L-Trp, which is a natural source of AhR ligands [139]. CD69 deficient mice showed reduced skin inflammation upon intradermal injection of rIL-23, and CD69 null dermal TCRγδ cells displayed reduced LAT1 expression, decreased L-Trp uptake, and AhR activation that resulted in reduced IL-22 production. Interestingly, IL-23 and IL-1β stimulation induced CD69 and LAT1 expression in Tγδ17 cells, providing a potential link between IL-23 signaling and AhR activation through LAT1-mediated uptake of L-Trp [139,140] (Figure 3). The pivotal role of AhR in the regulation of Th17 and Tγδ17 responses has attracted interest for therapeutic targeting. However, AhR targeting in a systemic manner may be challenging, as AhR activation exerts pro-inflammatory and anti-inflammatory roles in different tissues and cell types. For example, the AhR inhibitor, CH-223191, reduced inflammation parameters in a psoriasis model induced by intradermal injection of rIL-23 [139] (Table 1). In contrast, AhR-deficient mice developed exacerbated skin inflammation upon Imiquimod treatment, and AhR activation using the agonist FICZ ameliorated the pathology (Table 1). Moreover, AhR inhibition using CH-223191 increased inflammation parameters in skin biopsies from psoriatic patients [141]. In this work, the protective effects of FIZC were mediated by AhR activation in non-hematopoietic skin cells. AhR deletion in vascular endothelial cells also exacerbated disease parameters in both Imiquimod and subcutaneous rIL-23 models of skin inflammation [142]. Thus, AhR activation plays opposite roles in hematopoietic and non-hematopoietic cells in the context of skin inflammation. In the gut, AhR-dependent IL-22 production is required for maintenance of intestinal homeostasis [143] and for the protection against *Citrobacter rodentium* infection [33,144]. Thus, therapeutic targeting of AhR requires further investigation to determine the specific targets in different cell types, combined with the delivery of agonist or antagonist to the desired tissues or cells using, for example, nanoparticles [145]. The manipulation of cell-specific upstream regulators of AhR represents another option for therapeutic intervention. In this context, nitric oxide (NO)-inhibited AhR expression and Th17 differentiation in vitro and mice deficient for the inducible nitric oxide synthase (iNOS) developed exacerbated EAE with increased numbers of Th17 cells and enhanced expression of AhR [146]. In fact, some preclinical trials using nitric oxide carriers have shown protective effects in EAE [147] (Table 1). Thus, the manipulation of NO signaling could be of interest to target AhR in T cells [148].

## 5. IL-23-Regulated Transcription Factors Beyond STAT3: RORγt, Blimp, NF-κB, Tbet, Satb1, and GATA3

### 5.1. RORγt

The most studied transcription factor downstream the IL-23 signaling cascade beyond STAT3 is the retinoic acid receptor-related orphan receptor-γt (RORγt, encoded by *Rorc*), the master regulator of Th17 differentiation [149]. IL-23-mediated activation of STAT3 induces the expression of RORγt, which is crucial for IL-17 production [122,149,150] (Figure 4). Interestingly, not only Th17 cells but all IL-23-target populations are characterized by the expression of RORγt [149,151]. RORγt transcriptional activity is required for IL-23R expression [150,152], and hence, IL-23R closely mirrors RORγt expression in vivo (Immunological Genome Project [153]). Thus, the targeting of RORγt stands out as an opportunity for development of new treatments in IL-23-related diseases. To date, several RORγt antagonists and inverse agonists have been identified and tested in different models of inflammatory diseases [154]. The cardiac glycoside digoxin was found to inhibit RORγt binding to target loci, such as IL-17 and IL-23R in Th17 cells, and the administration of digoxin delayed the onset and reduced the clinical severity in the EAE model [155] (Table 1). The synthetic ligand SR1001, a reverse agonist of orphan nuclear receptors RORα and RORγt, blocked Th17 in vitro differentiation and had protective effects in EAE development [156] (Table 1). A screening using the RORγt ligand-binding domain identified the TMP778 compound as an inverse agonist with a more potent inhibitory effect on Th17 in vitro differentiation and EAE protection than digoxin [157] (Table 1). The impact of TMP778 on Th17 transcriptional profile analyzed by RNAseq determined that the most pronounced effect of RORγt inhibition was the decreased expression of Th17 cell signature genes, together with an increase in Th1 cell signature genes. The transcriptional analysis also revealed that TMP778 treatment closely mimicked RORγt deletion, and effects were mostly restricted to Th17 cells. This study identified the RORγt binding sites at Th17 genome, offering a wide network of novel potential targets for therapeutic intervention [157]. This work also reports an orally available compound, GSK805, with a more potent RORγt inhibitory effect than TMP778. Remarkably, the oral administration of GSK805 ameliorated the severity in the EAE model (Table 1). As mentioned above, oral delivery of treatments stands out as a potential improvement in patient quality of life compared to intravenous injections. In addition to Th17 cells, RORγt inhibitors are expected to have an impact in other RORγt-expressing cells. JNJ-54271074, another RORγt inverse agonist, inhibits IL-17 and IL-22 production in murine Th17, Tγδ, and NKT cells and, remarkably, in peripheral blood mononuclear cells (PBMC) obtained from rheumatoid arthritis patients [158] (Table 1). This compound ameliorated disease scores in the CIA model and in IL-23-induced skin inflammation models in a dose-dependent manner. The BIX119 compound, a RORγt agonist, blocks human Th17 differentiation and also inhibits IL-17 production but not IL-22 in innate-like populations, such as human Tγδ17 and iNKT17 cells in vitro [159]. Thus, the screening for RORγt inhibitors is a very active field of research. In addition to pharmacological compounds, bile acid metabolites [160] and intermediates of the cholesterol biosynthetic pathway [161] have been found to inhibit RORγt, opening new avenues to manipulate the effector functions of RORγt-expressing cells.

### 5.2. Blimp-1

IL-23 primarily activates STAT3, which maintains RORγt expression [150]. However, STAT3 is activated downstream many other cytokines, such as IL-6 and IL-21, and thus, STAT3 alone does not fully explain the pathogenic actions of IL-23. Other transcription factors such as Blimp-1 (encoded by *Prdm1*) have been shown to participate in the IL-23 signaling cascade [162] (Figure 4). Blimp-1 is a key regulator of B cell differentiation to antibody secreting plasma cells [163], and it also has critical roles in the regulation of effector T cells [164]. IL-23 stimulation of Th17 induces Blimp-1 expression in a STAT3-dependent manner. Blimp-1-deficient Th17 were not able to co-express GM-CSF and interferon gamma (IFN-γ), αγδ mice with conditional ablation of Blimp-1 using the Lck distal promoter-Cre system showed reduced EAE development [162]. Mechanistically, Chip-Seq analysis found Blimp-1 bound to *Il23r*, *Il17*, and *Csf2* (that encodes for GM-CSF) regulatory regions, and Blimp-1 overexpression was sufficient to induce GM-CSF production in Th17 cells in the absence of IL-23. Thus, Blimp-1 can regulate Th17 effector function by amplifying IL-23 signaling (induction of IL-23R expression) and enhancing GM-CSF transcription. However, using a different knockout model for Blimp-1 expression in T cells, other work found an uncontrolled Th17 cell-driven central nervous system pathology in Blimp-1 deficient mice [165]. This controversy may reflect the fact that Blimp-1 seems to play distinct roles during the priming and the effector phase of EAE, as other work found delayed EAE onset but enhanced disease severity during the effector phase in Blimp-1 conditional knockout mice [166].

### 5.3. NF-κB

Studies in the context of osteoclastogenesis, bone destruction, and arthritis pathology have identified a relevant role of the IL-23/IL17 axis in these processes, and they have linked IL-23 signaling with the activation of the transcription factor NF-κB. In these pathological conditions, excessive generation of osteoclast degrade bone matrix and the receptor activator of NF-κB ligand (RANKL) is a key osteoclastogenic cytokine expressed by mesenchymal cells but also by T cells [167] (Figure 4). IL-23 and IL-17a deficient animals displayed reduced bone destruction and osteoclast formation in a LPS-induced model of inflammatory bone destruction [168]. Moreover, in an animal model of spontaneous arthritis (IL-1Ra deficient mice), IL-23 signaling induced the expression of RANKL both in CD4 T cells and synoviocytes [169,170,171]. IL-23-induced expression of RANKL was abolished by treatment with Jak2 inhibitor AG490 and, interestingly, by the NF-κB inhibitor parthenolide, suggesting that IL-23 signaling pathway triggers the activation of NF-κB [170]. Mechanistically, some studies have shown that IL-23 promotes the phosphorylation of the NF-κB inhibitor alpha (IκBα) [169,172]. IκBα phosphorylation promotes its degradation, allowing NF-κB translocation to the nucleus [173]. Thus, although further studies are required to determine the role of IL-23-induced NF-κB activation, these data suggest that IL-23 promotes the canonical pathway of NF-κB activation. The inhibition of NF-κB is of great interest in the context of inflammatory diseases, not only due to the potential interference with IL-23 signaling cascade but also because this transcription factor is involved in several inflammatory responses. NF-κB promotes the transcription of a wide range of pro-inflammatory genes: cytokines, chemokines, adhesion molecules, and anti-apoptotic factors, and it is a central mediator of NLRP3 inflammasome activation (recently reviewed in [174,175]). Given the key role of NF-κB in immunity, it has been implicated in several inflammatory diseases and considered as a therapeutic target. Several inhibitors have been developed to interfere with NF-κB activation at different levels. Different inhibitors aim to prevent IκBα phosphorylation and further degradation and, for example, well-known anti-inflammatory drugs, such aspirin or salicylate are able to inhibit the IκB kinase (IKK) and prevent IκBα phosphorylation [176]. The proteasome inhibitor Bortezomib (Velcade) is able to block IκBα degradation, a key step for NF-κB release [177]. There are also inhibitors that block NF-κB nuclear translocation and DNA binding. However, due to the pivotal role of NF-κB both in normal and pathogenic immune response, its inhibition may cause severe side effects. NF-κB controls key cell processes, such as cell survival, differentiation, and proliferation, that are required for the development and activation of immune cells, for the skeletal system, and epithelium. Aberrant NF-κB activation leads to dysregulation of these same processes and contributes to the development of cancer and chronic inflammatory diseases [178]. NF-κB is activated by a plethora of immune receptors involved in both innate and adaptive immune response (i.e., Toll-like receptors, TNFα, BCR, TCR), and its activation leads to the transcription of pro-inflammatory cytokines, chemokines, cell adhesion molecules, factors of the complement cascade, and acute phase proteins. Thus, it is easy to understand how the excessive activation of NF-κB pathway contributes to chronic inflammatory diseases. At the same time, mutations in components of the NF-κB pathway result in a variety of disease susceptibilities that provide clues about the role of NF-κB in steady state [175]. Mutations in IκBα, a key NF-κB upstream regulator, lead to poor T cell responses to antigen stimuli, lack of germinal centers, and decreased responses to Toll-like receptors or TNFα [179]. Hypomorphic mutations in NEMO, another NF-κB upstream regulator, cause skin inflammation related disorders, suggesting a key role of NF-κB in keratinocyte function [180,181]. Defects in NEMO [182], the ubiquitin ligase A20 (TNFAIP3) [183], and RELA haploinsufficiency [184] lead to intestinal inflammation, suggesting a relevant role of the NF-κB signaling pathway to maintain intestinal epithelium integrity and host defense. Thus, a better understanding of the molecular mechanisms implicated in NF-κB activation in individual pathologies and identification of specific IL-23/NF-κB-regulated genes is key for the development of specific therapeutic strategies to interfere with NF-κB in the context of IL-23-related pathologies.

### 5.4. T-bet

The expression of the transcription factor T-bet (encoded by *Tbx21*) was identified as part of the pathogenic signature induced by IL-23 in Th17 cells and required for the co-expression of IL-17 and IFNγ [19,185]. After some initial controversy, it has been established that T-bet expression is not essential for EAE development [186,187]. IL-23 does not induce T-bet expression, although T-bet may be required for optimal production of IL-17 in response to IL-23 in Th17 cells, at least in vitro [188]. In contrast, a different scenario has been found in intestinal T cell responses. In a T cell transfer colitis model, T-bet deficient T cells promoted an exacerbated Th17 response in the gut, and T-bet-deficient CD4 T cells isolated from an inflamed colon were hyper-responsive to IL-23 in terms of STAT3 phosphorylation and IL-17 and IL-22 production [189]. In addition, IL-23-mediated IL-22 production play a protective role against *Citrobacter rodentium* infection, and the protective IL-22 response required T-bet expression [144]. Thus, the relationship between IL-23 and T-bet requires further investigations before considering the manipulation of T-bet as a potential therapeutic opportunity.

### 5.5. Satb1

Recently, IL-23 was shown to induce the expression of the special AT-rich sequence-binding protein-1 (Satb1) [190] (Figure 4). Satb1 is a chromatin organizer with an essential role controlling the expression of a large number of genes that participate in T cell development and activation, and Satb1 directly recruits chromatin modifiers to the loci of its target genes [191]. Mice with conditional deletion of Satb1 in Th17 cells were resistant to the development of EAE. Interestingly, Satb1-deficient Th17 cells showed normal IL-17 but impaired GM-CSF production [190], a key cytokine for EAE progression [7,8]. IL-23 but not IL-6 or IL-1β induced Satb1 expression. Mechanistically, RNAseq analysis of wild type and Satb1-deficient Th17 cells extracted from the spinal cord after EAE induction showed that the expression of basic helix-loop-helix transcription factor Bhlhe40 was lost in Satb1-null cells [190]. The transcription factor Bhlhe40 is required for GM-CSF production and EAE induction [192,193], and the reconstitution of Bhlhe40 expression by retroviral transduction restored the pathogenic function of Sabt1-deficient Th17 [190]. Thus, the targeting of the Satb1/Bhlhe40 axis would be an interesting therapeutic opportunity, in particular for multiple sclerosis treatments. In this context, Sabt1 is regulated by different post-transcriptional modifications that exert opposing roles on its DNA binding capacity. For example, Satb1 phosphorylation by protein kinase C (PKC) increases DNA binding affinity, while acetylation by the K acetyltransferase 2B (PCAF/KAT2B) reduces DNA binding [194]. Thus, although the hypothesis needs to be experimentally tested, inhibition of PKC activity can potentially decrease GM-CSF production.

### 5.6. GATA3 Inhibition

In addition to the production of pro-inflammatory cytokines, IL-23 promotes intestinal inflammation through the inhibition of regulatory T cell (Tregs) differentiation [195,196]. In contrast, the alarmin IL-33 induced *Foxp3* expression though activation and recruitment of GATA3 to the *Foxp3* promoter and the *St2* enhancer in colonic Tregs [197]. *St2*, also known as *Il1rl1*, encodes for the IL-33 receptor. Notably, IL-23 inhibited the induction of *Gata3* and *St2* mRNA in Treg differentiation conditions, and further stimulation of these cells with IL-33 failed to recruit GATA3 to the *St2* enhancer (Figure 4). Interestingly, sorted *St2*^+^ Treg cells from the colon expressed detectable levels of *Il23r* mRNA, and treatment of TCR-activated thymus-derived Treg cells with IL-23 impaired IL-33-mediated GATA3 phosphorylation and abrogated the expression of genes co-regulated by Foxp3 and GATA3 upon IL-33 stimulation. Moreover, the addition of a specific inhibitor of STAT3 reversed the inhibitory effect of IL-23, suggesting that IL-23 inhibits IL-33 through STAT3 activation. Different studies have shown a protective role for IL-33 in intestinal mucosa [197,198] and psoriasis [199]. Further studies characterizing the molecular mechanisms underlying IL-23 inhibition of Tregs responsiveness to IL-33 can reveal new therapeutic strategies to counteract the pathogenic effects of IL-23.

## 6. IL-23 and IL-1β: Signaling Crosstalk

Early studies on IL-23 pathogenic function identified a synergistic behavior with IL-1β. IL-1β is synthetized as an inactive precursor and processed by caspase-1 upon inflammasome activation. IL-1β binding to its surface receptor triggers a signaling cascade mediated by the myeloid differentiation primary response 88 (MyD88) adaptor protein, IL-1R-associated kinases (IRAKs), and TRAF6 [200] (Figure 5). IL-1β plays a pivotal role in the orchestration of both innate and adaptive immune responses [201]. Stimulation with IL-23 and IL-1β greatly increases the production of pro-inflammatory cytokines, such as IL-17, IL-22, or GM-CSF, compared with the effect of both cytokines alone in Tγδ17 and Th17 cells [202,203,204,205], and the specific deletion of MyD88 expression in T cells abrogated IL-23 and IL-1β synergy [206]. Thus, targeting the MyD88/IRAKs/TRAF6 signaling cascade holds promise for novel therapeutic approaches. Different inhibitors targeting IRAK4 are currently progressing to clinical trials [48,207]. In addition, IL-1R8 (single Ig IL-1-related receptor, SIGIRR), a negative regulator of IL-1 signaling [201], interfered with the activation of the MyD88/IRAK4/IRAK1 signaling module, and SIGIRR-deficient animals showed increased susceptibility to EAE development [208]. Thus, therapeutic strategies aimed to increase the expression of SIGIRR may be protective in the context of IL-23-related inflammatory diseases.

The molecular mechanisms underlying the IL-23/IL-1β synergy are not completely defined, but positive feedback loops have been shown at different levels. A key feedback mechanism is provided by IL-23-mediated induction of the expression of IL-1 receptor (IL-1R1). IL-23 stimulation induced the expression of IL-1R1 in IL-17-producing CD4 cells [5] and intestinal CD4+ T cells and ILCs [203]. Conversely, IL-1β signaling is also required for IL-23R expression in Th17 cells, as both IL-1R and MyD88 deficient Th17 cells showed decreased expression of IL-23R [206,209]. Another positive feedback loop of IL-1β on IL-23 signaling cascade is the regulation of SOCS3 expression. As mentioned above, SOCS3 is a negative regulator of STAT3 activation (Figure 2). Work on Th17 cells showed that the combination of IL-1β plus IL-23 induced an increased and more sustained pSTAT3-Y705 than IL-23 alone. IL-1β stimulation decreased the expression of *Socs3* mRNA induced by IL-23. Hence, the reduced Socs3 expression might be responsible for the increased and sustained STAT3 phosphorylation [210] (Figure 5). Pharmacological inhibition of NF-κB transcriptional activity abolished the IL-1β-reduced expression of Socs3 and, accordingly, the enhanced pSTAT3-Y705 in the presence of IL-23/IL-1β was abolished in NF-κB/RelA-deficient Th17 cells [210]. Mechanistically, IL-1β-induced RelA activation increased the binding of IL-23-induced STAT3 to regulatory elements in the *Il17a/f* locus, providing a relevant mechanism for IL-23/IL-1β synergy [211].

Another level of IL-23/IL-1β signaling crosstalk has been established through the activation of mTORC1. In Th17 differentiation cultures, IL-23 and IL-1β triggered the activation of mTORC1, and the treatment with mTORC1 inhibitor rapamycin reduced IL-17 production [206]. In addition to Th17 and Tγδ17, IL-23R is also expressed in specific subpopulations of neutrophils [212], and IL-23 stimulation increased IL-17 and IL-22 production and the expression of RORγt and AhR in peritoneal neutrophils [213]. IL-23 induced mTORC1 activation, and IL-23-induced mTORC1 activation was abolished by rapamycin and AZD8055, an mTORC1/C2 inhibitor. More importantly, both inhibitors suppressed the IL-17 and IL-22 production induced by IL-23. Mechanistically, both rapamycin and AZD8055 decreased the IL-23-induced expression of RORγt and AhR (Figure 5). To explore the upstream signaling pathway regulating IL-23/mTORC1 activation, the PI3K antagonist LY294002 was shown to inhibit IL-23-induced mTORC1 activation and IL-17/IL-22 production. However, data obtained using the PI3K inhibitors LY294002 or worthmannin must be carefully considered, as both have off-target effects and they directly inhibit mTORC1 and the Pim family of kinases [214].

In dermal Tγδ cells, IL-23 and IL-1β not only promote the increased production of IL-17 but also induced cell proliferation, and rapamycin treatment significantly decreased proliferation of dermal Tγδ cell in vitro and IL-17 production upon IL-1β/IL-23 stimulation. IL-1β, but not IL-23, promoted mTORC1 activation in dermal Tγδ cells [215]. Although rapamycin treatment primarily affects mTORC1 function, contradictory effects have been reported regarding the impact of prolonged or high doses of rapamycin on the activity of mTORC2 [216,217]. Specific deletion of *Raptor* (mTORC1 subunit) and *Rictor* (mTORC2 subunit) using the CD2-Cre system showed that mTORC1 activity was required for IL-1β and IL-23/IL-1β-induced dermal Tγδ cells proliferation, but interestingly, IL-17 production was not affected by *Raptor* deletion. In contrast, both proliferation and IL-17 production were reduced in *Rictor*-deficient dermal Tγδ cells [215]. Mechanistically, this work showed that the combined action of IL-23 and IL-1β greatly induced the expression of the interferon regulatory factor 4 transcription factor (IRF4), a key transcription factor involved in Th17 differentiation [209,218]. IRF4 induction upon IL-23/IL-1β was abolished in *Rictor*-deficient but not in *Raptor*-deficient Tγδ cells, and siRNA against IRF4 reduced IL-17 production from dermal Tγδ cells [215]. Thus, mTORC2 play a key in IL-23/IL-1β synergy through IRF4 induction in skin resident Tγδ cells (Figure 5). An additional mechanism reported for IL-23/IL-1β synergistic action on mTORC1 activation is the up-regulation of the expression of the amino acid transporter LAT1 (*Slc7a5*) in Tγδ17 and Th17 murine cells [140]. IL-23/IL-1β mTORC1 activation and Tγδ17 cell proliferation were abolished in Tγδ17 cells obtained from LAT1-deficient mice (*Slc7a5*^fl/fl^xRORγt-Cre mice), suggesting that mTORC1 activation downstream IL-23/IL-1β requires LAT1 expression to promote pathogenic effector function in T cells. Accordingly, mice with specific deletion of LAT1 in RORγt-expressing cells were partially protected for IMQ-induced skin inflammation, and both pharmacological inhibition of LAT1 using the JPH203 inhibitor and rapamycin administration ameliorated IMQ-induced psoriasis severity (Table 1). Interestingly, the protective effects of LAT1 inhibition were independent of AhR function, as JPH203 still inhibited Tγδ17 cell proliferation in AhR-deficient mice [140] (Figure 5). Thus, in addition to rapamycin treatment, the inhibition of mTORC1 by LAT1 targeting is emerging as a potential therapy for cancer treatment [219], and these therapeutic strategies may be efficient to interfere with an IL-23/IL-1β signaling cascade in inflammatory diseases.

**Table 1 cells-09-02044-t001:** Selected small molecule inhibitors, drugs, and protein scaffolds that interfere with IL-23 signaling cascade.

Inhibitor	Target	Clinical Phase	Disease/Model	Route	Reference
IL-23R peptide antagonist	IL-23R	Preclinical	rIL-23-induced skin inflammation model, anti-CD40 systemic inflammation, CIA	i.p.*	[63]
IL-23 Alphabodies	IL-23	Preclinical	rIL-23-induced skin inflammation model	i.d., i.p.	[64]
Tofacitinib	Jak1, Jak3, Jak2	Approved	Rheumatoid and psoriasic arthritis, ulcerative colitis	oral	[78]
		Phase 3	Juvenile arthritis, systemic lupus erythematosus	oral	
		Phase 2	Atopic dermatitis	topical	
Ruxolitinib	Jak1, Jak2	Phase 2	Rheumatoid arthritis	oral	[78]
		Phase 2	Atopic dermatitis	topical	
Baricitinib	Jak1, Jak2	Approved	Rheumatoid arthritis	oral	[78]
		Phase 2	Psoriasis, systemic lupus erythematosus	oral	
		Phase 3	Atopic dermatitis	topical	
BMS-986165	Tyk2	Phase 2	Psoriasis, rheumatoid arthritis, Crohn’s disease, ulcerative colitis, systemic lupus erythematosus	oral	[80]
Brepocitinib (PF-06700841)	Tyk2, Jak1	Phase 2	Psoriasis, Crohn’s disease, ulcerative colitis	oral	[81]
Baicalin	STAT3/SOCS3	Preclinical	EAE model	i.p.	[98]
Fasudil	ROCK	Preclinical	EAE model	oral, i.p.	[116,117]
WAR5 compound	ROCK	Preclinical	EAE model	i.p.	[118]
Statins	ROCK	Preclinical	EAE model	i.p.	[119]
KD025	ROCK	Phase 2	Psoriasis vulgaris	oral	[120]
Fingolimod (FTY720)	S1PR1	Approved	Multiple sclerosis	oral	[127]
		Preclinical	IMQ-induced skin inflammation model	i.p.	[128]
Natalizumab	VLA4	Approved	Multiple sclerosis, Crohn’s disease	i.v.	[129]
TEPP-46	PKM2	Preclinical	EAE model	i.p.	[132]
CH-223191	AhR inhibitor	Preclinical	IL-23-induced skin inflammation model	i.p.	[139]
FICZ	AhR agonist	Preclinical	IMQ-induced skin inflammation model	i.p	[141]
Nitric oxide carriers	AhR	Preclinical	EAE model	oral	[147]
Digoxin	RORγt	Preclinical	EAE model	i.p.	[155]
SR1001	RORαt, RORγt	Preclinical	EAE model	i.p.	[156]
TMP778	RORγt	Preclinical	EAE model	s.c.	[157]
GSK805	RORγt	Preclinical	EAE model	s.c.	[157]
JNJ-54271074	RORγt	Preclinical	CIA and IL-23-induced skin inflammation models	oral	[158]
JPH203	LAT1	Preclinical	IMQ-induced skin inflammation model	i.p.	[140]
Rapamycin	mTOR	Preclinical	IMQ-induced skin inflammation model	i.p.	[140]

Abbreviations: rIL-23, recombinant IL-23; i.p, intraperitoneal; i.d., intradermal: i.v., intravenous; s.c. subcutaneous.

## 7. Concluding Remarks

Here, we have collected information about specific molecular events that participate in IL-23 signaling cascade and how this knowledge can be taken further into the development of therapeutic intervention using small molecule inhibitors. Despite the success of neutralizing antibodies to interfere with secreted mediators, such as IL-23 and IL-17, not all patients show a full response, and their averaged cost prevent their use as a first line of treatment. In comparison, small molecule inhibitors present advantageous features for treatment of IL-23-mediated diseases. Monoclonal antibodies can target only secreted mediators or cell surface proteins, while small molecule inhibitors can be directed to intracellular targets. Antibodies are intravenously administered, while small molecule inhibitors can be formulated for oral or even topical administration. Oral administration of drugs can have a great impact in patient quality of life. Topical formulas are of particular interest for skin conditions, as the affected zone is accessible for topical administration of drugs. Another example is multiple sclerosis, where the hematoencephalic barrier may represent a difficulty for monoclonal antibodies due to their large molecular weight, while small molecule inhibitors can be delivered to brain tissues. In summary, small molecule inhibitors, used alone or in combination with antibody treatments, can results in improved treatment efficacy.

In addition to inflammatory diseases, IL-23 and IL-23 target populations have been implicated in cancer development [220]. For example, IL-23p19-deficient mice are resistant to tumor formation in a cutaneous chemical carcinogenesis model [221,222] and in colorectal cancer mouse models [223]. IL-23 has pro-tumor function in intestinal adenomas [224], lung metastases [225], and prostate cancer [226]. There is also evidence that IL-23 has a role in Alzheimer´s disease, as amyloidosis-prone *APPPS1* mice lacking IL-23 and IL-12 have a decreased in amyloid-β plaque burden, and the administration of p40 neutralizing antibodies reduced the pathology [227]. Thus, therapeutic strategies to interfere with IL-23 signaling cascade could be useful for pathological conditions.

Finally, although most of the research about IL-23 is focused on its role in inflammatory and autoimmune diseases, it is important to mention that Th17 and other IL-17-producing cells did not evolve to cause autoimmunity and other inflammatory diseases but to provide effective host defense against extracellular pathogens. In Tγδ17 cells, IL-23 is required not only to induce pathogenic function but also to promote Tγδ17 differentiation from TCRγδ progenitors in the periphery [228,229]. Thus, IL-23 function is required to maintain Tγδ17 protective functions. IL-23 is required for protection against *Citrobacter rododentium* [121,122], *Listeria monocytogenes* [230], and *Klebsiella pneumonia* [231]. The IL-23/IL-17 axis is essential in the resolution of infection by *Candida albicans* [232,233], *Staphylococcus aureus* [234], and *Bordetella pertussis* [235], and IL-23-induced IL-22 production is involved in the clearance of lymphocytic choriomeningitis virus (LCMV) infection [236]. Thus, the study of the molecular mechanisms downstream IL-23 is not only relevant to block its pathogenic actions, but also to boost protective function against specific pathogens that could be of special relevance in the context of immunosuppressed individuals.

## Figures and Tables

**Figure 1 cells-09-02044-f001:**
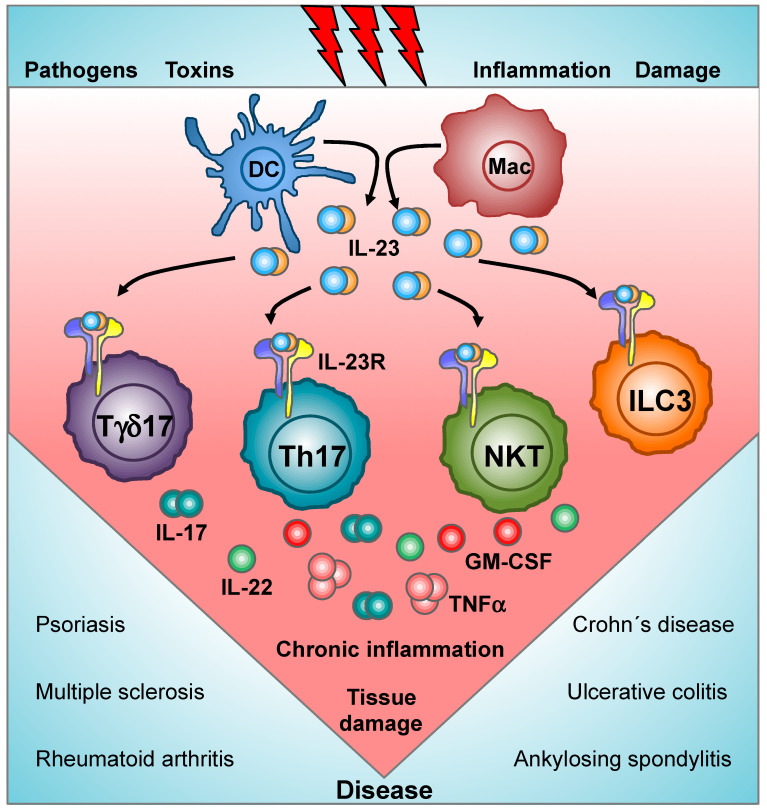
Role of interleukin 23 (IL-23) in chronic inflammatory diseases. IL-23 is produced by dendritic cells (DCs) and activated macrophages, and its actions are mainly mediated by the CD4 T helper subset Th17, a distinct subpopulation of gamma/delta T cells (Tγδ17 cells), subsets of natural killer T (NKT) cells, and type 3 innate lymphoid cells (ILC3s). IL-23 signaling promotes the production of pro-inflammatory mediators (IL-17, IL-22, granulocyte-macrophage colony-stimulating (GM-CSF), or the tumor necrosis factor (TNFα)) by target populations. These pathogenic mediators promote the recruitment and activation of granulocytes and macrophages, causing the tissue damage that induce chronic inflammation and, finally, the development of clinical symptoms.

**Figure 2 cells-09-02044-f002:**
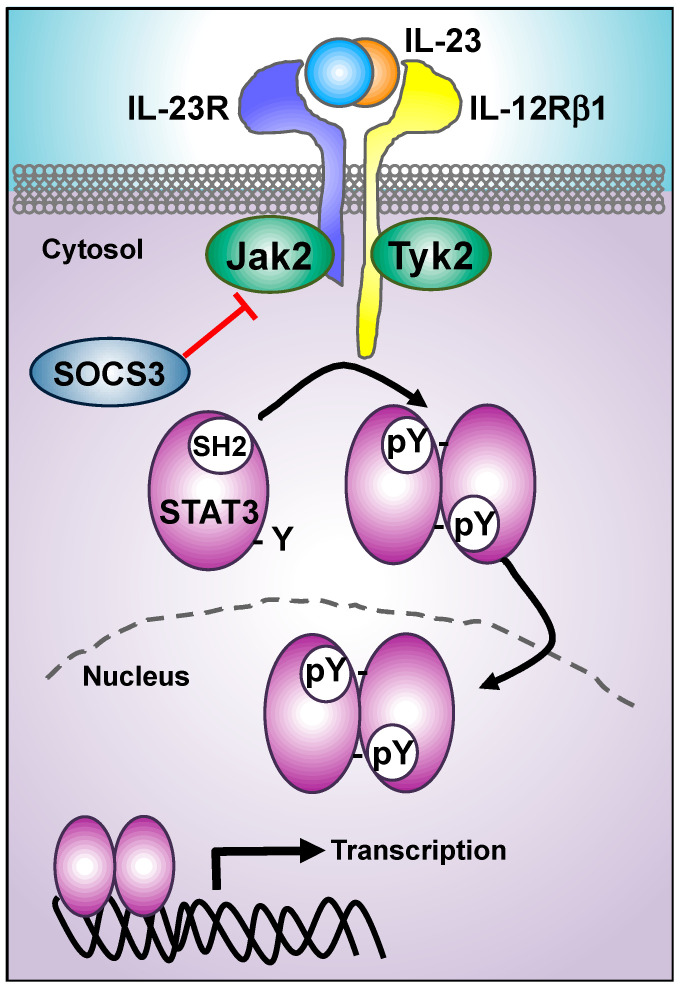
IL-23 proximal signaling cascade: Janus tyrosine kinases (JAKs)/signal transducer and activator of transcription (STATs) module. IL-23 receptor complex is associated with the JAK family members Jak2 and Tyk2. IL-23 binding to the IL-23R promotes JAKs activation and phosphorylation of Tyr residues (pY) in both the JAKs themselves and in the IL-23R cytoplasmic tail, creating docking sites for STAT3 monomers through Src homology 2 domains (SH2). Active Jak2/Tyk2 phosphorylate STAT3 monomers, leading to dimerization, nuclear translocation, and DNA binding to target gene promoters.

**Figure 3 cells-09-02044-f003:**
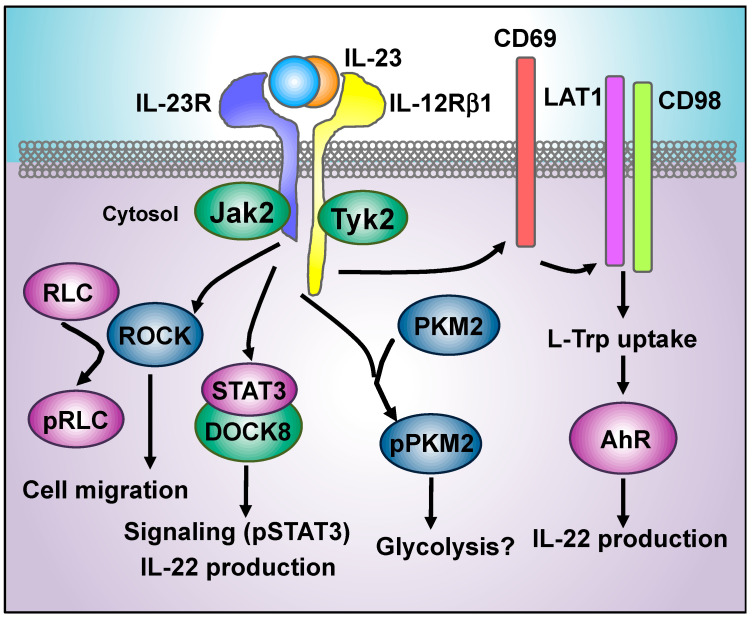
New players in IL-23 proximal signaling events. IL-23 promotes phosphorylation of the myosin regulatory light chain (RLC) via Jak2 and ROCK activation, and IL-23 induces migration of Tγδ17 and Th17 cells in a ROCK dependent manner. The association of the Dedicator of cytokinesis 8 (DOCK8) to STAT3 is required for sustained STAT3 phosphorylation and IL-22 production downstream IL-23. IL-23 triggered the phosphorylation of pyruvate kinase isoform PKM2-Ser37 residue, promoted its nuclear translocation, and induced the expression of PKM2 downstream target genes. IL-23 induces CD69 expression, and CD69 associates with LAT1/CD98 amino acid transporter to promote L-Trp uptake. Derivatives from L-Trp metabolism activate AhR to induce IL-22 production in Tγδ17 cells.

**Figure 4 cells-09-02044-f004:**
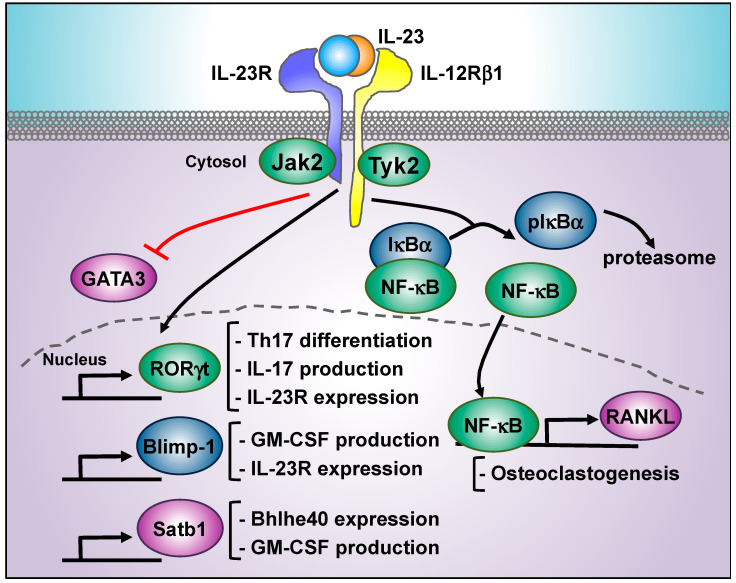
IL-23-regulated transcription factors. IL-23 induces the expression of the transcription factor retinoic acid receptor-related orphan receptor-γt (RORγt), master regulator of Th17 cell differentiation and essential for IL-17 production and IL-23R expression. IL-23 promotes the expression of the transcription factor Blimp1, required for GM-CSF production and IL-23R expression in Th17 cells. IL-23 stimulation increases the expression of the chromatin organizer Satb1 that in turn promotes Bhlhe40 expression that are required for GM-CSF production. IL-23 promotes IκBα phosphorylation and proteasomal degradation, allowing NF-κB translocation to the nucleus and the induction of the receptor activator of NF-κB ligand (RANKL) expression, a key osteoclastogenic cytokine involved in bone destruction and arthritis pathology.

**Figure 5 cells-09-02044-f005:**
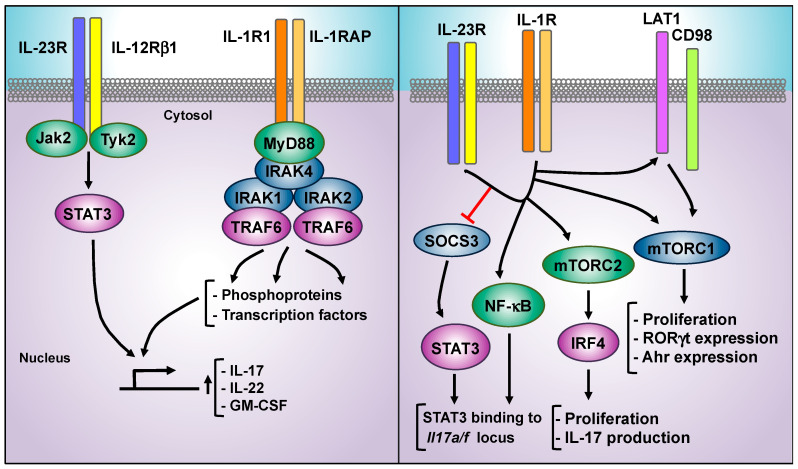
IL-23 and IL-1β: signaling crosstalk. The combined action of IL-23 and IL-1β greatly increases the production of IL-17, IL-22, and GM-CSF. *Right*, IL-23 and IL-1β signaling cascades at a glance: IL-23 signaling cascade is mediated by the JAK/STAT module, while IL-1β signal is transduced by MyD88/IRAKs/TRAF6 module. MyD88/IRAKs/TRAF6 induce the activation of several phosphoproteins and transcription factors. *Left*, IL-1β stimulation decreases IL-23-induced expression of Socs3 in an NF-κB-dependent manner, and decreased Socs3 expression upon IL-23/IL-1β stimulation leads to an increased STAT3 phosphorylation. IL-1β-induced NF-κB/RelA activation increased the binding of IL-23-induced STAT3 to regulatory elements in the *Il17a/f* locus. IL-23 and IL-1β promote mTORC1 and mTORC2 activation. IL-23/IL-1β activation of mTORC2 induces the expression of the transcription factor IRF4, required for IL-17 production and Tγδ cell proliferation. IL-23/IL-1β-induced mTORC1 activation promotes Tγδ cell proliferation and induces RORγt and AhR expression in neutrophils. IL-23/IL-1β co-stimulation increases the expression of LAT1 and induces the activation of mTORC1 and the proliferation in Tγδ17 cells.

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
