# Peer review of "Decoding IL-23 Signaling Cascade for New Therapeutic Opportunities"

_cells, 2020, doi:10.3390/cells9092044_

Round 1

Reviewer 1 Report

The review focused on IL-23 signaling and opportunities for the therapy of IL-23-mediated inflammatory diseases. After an introduction into IL-23 the authors shortly described the IL-23 signaling complex. Then, they concentrate on the JAK/STAT pathway. They also introduced new players in IL-23 signaling events and highlighted IL-23 regulated transcription factors. Additionally, Pastor-Fernández and colleagues summarized the connection between IL-23 signaling pathway and the aryl hydrocarbon receptor and the signaling crosstalk between IL-23 and IL-1β. The authors finalized their review article with concluding remarks regarding other IL-23-mediated diseases and IL-23-mediated host defense against extracellular pathogens.

This review article is clearly and well written; and the different sections are in a logical order. The authors give an excellent overview about IL-23 and signaling events connected to inflammatory diseases. New therapeutic opportunities are mentioned and explained in detail.

However, the paper requires some minor changes prior to publication:

Line 36: The authors did not mention IL-39, which was identified in mice. Up to now, IL-39 could not be found in humans.

Figure 1: Greek letters in gamma/delta T cells and TNF alpha are not shown.

Figure 2 and 3: Greek letter in IL-12Rbeta1 is not shown.

Figure 4: Greek letter in IL-12Rbeta1 is not shown. Glycolisis should be Glycolysis.

Figure 5: Greek letters in IL-12Rbeta1, RORgammat, NF-kappaB, IkappaBalpha are not shown.

Line 390: There is too much space between IFN-gamma, and mice….

Figure 6: Greek letters in IL-12Rbeta1, NF-kappaB, IkappaBalpha and IKKalpha/beta are not shown.

Line 574: There is too mice space between IL-1beta and the….

Author Response

We greatly appreciate reviewer´s comments and support for the publication of our manuscript, and we have addressed the minor changes requested.

Minor changes

- Regarding the IL-39 cytokine, we agree with the reviewer that this cytokine should be mentioned in the context of an IL-23 review and accordingly, IL-39 has been included in lines 38 and 212-214.

- Regarding the missing Greek lettering in the figures, it seems that there has been a problem during conversion to pdf format during submission, probably due to some kind of software incompatibility. Figures have been reinserted in a different format to fix the problem. Otherwise, these errors will be fixed during the technical revision of the manuscript.

- Other minor typing errors like excessive spacing have been corrected

Reviewer 2 Report

Evaluation of the review article „Decoding IL-23 signaling cascade for new therapeutic opportunities“.

Inhibition of IL-23 receptor signaling function is a promising therapeutic approach for treatment of  autoimmune diseases including psoriasis, psoriatic arthritis, Crohn’s disease and ulcerative colitis, rheumatoid arthritis or multiple sclerosis. Several approaches have already been tested to block the interaction between IL-23/p19 subunit with its cognate receptor by inhibitory monoclonal antibodies, artificial protein binders derived from small protein domain scaffolds, by soluble IL-23 receptor, in silico modified p19/IL-23 protein as well as small synthetic molecules, short peptides or their derivatives. Until recently, molecular structure of IL-23/IL23R has not been available and designing of efficient inhibitors of IL-23-mediated signaling was, therefore, challenging. Recently published structure of IL-23 cytokine in complex with IL-23 receptor by Bloch et al. opens new possibilities for designing more efficient IL-23 receptor antagonists that are essential for anti-inflammatory drug development. However, targeting particular intracellular members of signaling cascade represents a promising alternative.

In the submitted review, Pastor-Fernandez, Mariblanca and Navarro summarized overview of results related to blocking of IL-23 signaling cascade with emphasis on the role of small synthetic molecules as specific inhibitors of particular members of IL-23 receptor-related signaling cascade. The review is composed of an introductory section, six chapters and concluding remarks, with 6 inserted figures. My opinion is that in terms of the scientific content, review article fulfil criteria to be accepted, but I have several comments how to further improve the manuscript.

I address authors several following comments:

Major comments:

  1. While the chapters 2-to-5 are very well prepared and fulfil criteria to be expected from such full-length review, the introduction should be improved. I miss a molecular description of the all IL-23/IL-17 signaling cascade, yet I understand that authors want to focus only on IL-23 signaling. But it is imposible to further discuss roles of IL-22 and IL-17 in context to human autoimmune diseases and K-O mouse models without elementary description of the interaction of these key players of the IL-23 inflammatory cascade with their cognate receptor IL-22R1, and IL-17A/F with IL-17RA and IL-17RC. This should be included in the introduction.
  2. In the introiductory part, authors desribe strategies for therapeutic intervention in IL-23 signaling cascade. They mention blocking monoclonal antibodies as one strategy and small molecules as inhibitors as strategy 2. This should be completed by the insertion of strategy 3 describing development of scaffold proteins blocking IL-23/IL-17 cascade as non-Ig alternative to antibodes (for instance see Gebauer and Skerra, Engineered Protein Scaffolds as Next-Generation Therapeutics, Annual Review of Pharmacology and Toxicology, 2020). Till now, there are only several scaffold proteins as blockers of IL-23 signaling and those might be mentioned (Alphabody-IL-23 (Desmet 2014), ABD-derived IL-23 receptor blocking REX ligands (Kuchar 2014) and IL-23 blocking ILP blockers (Krizova 2017), IL-17-blocking Fynomer or IL-17 receptor A blockers (Hlavnickova 2018).
  3. While sections 2-6 are prepared carefully with many details of function of small inhibitors and roles of particular members of signaling cascade, I feel that it will be necessary to summarize these small molecule blockers/drugs into one big table indicating corresponding target molecules, clinical phase, drug for disease (f.i. psoriasis), administration (oral, topical, subcutaneous, intradermal, intraperitoneal, …) and corresponding reference. This would significantly increase impact of this review and bring better overview for readers for all described potential drugs.
  4. Chapter 6 describing role of AhR is very detailed and long in comparison to other parts of the review. This suggests that authors ascribe this part a substantial role in the manuscript. But the role of AhR is very complicated and controversial in terms of a therapeutical intervention and is closely related to ambiguity of IL-22 cytokine function, especially in IBD. If this is so important for this review, I would suggest to include also description of IL-22 function in psoriasis (stimulation of chemokines, controlling massive homing of granulocytes forming skin infiltrate) and in intestinal inflammation by IL-22 function control through a soluble IL22 binding protein (IL-22BP), thus tiny balancing between beneficial intestinal epithelial cell lining regeneration and deleterious pro-inflammatory effect in ulcerative colitis and also in colon carcinoma development. This might be discussed in context of possible targeting by small molecules, natural products or antibodies. If the role of IL-22/IL-22R1 interaction is here missing then I do not understand why this chapter should be so long and complex without further IL-22 context related to human diseases and possible therapeutic intervention.
  5. Signaling crosstalk between IL-23 and IL-1β forms a special subsection of this review. I think that this is very complex topic which deserves to be separated from this review and should be discussed in more detail in some other review. Maybe, more interesting for readers would be to shorten this section to focus only on the most important recent findings or to completely substitute this part with a chapter describing, instead, blocking of IL-17 cytokine and IL-17 receptor with emphasis on inhibitors of IL-17RA-mediated TRAF-6-related signaling cascade, yet authors declare to focus only on proximal signaling events. I miss the context of signaling inhibitors for IL-22 and IL-17-cascades in this otherwise very complex review.
  6. I am disappointed by general graphical representation used in this review. For instance, Figure 2 brings very low value, if authors want to demonstrate all players of IL-23 signaling complex, particular structures from PDB database should be downloaded to demonstrate key interacting domains. If p19 should be shown as four-helix bundle, this should not be shown as a linear protein. In general, Figures or rather schemes in the presented version seem to be a weakness of the manuscript.

Minor comments:

Correct typing errors in almost all figures.

Row 311 …inhibits egress of T cells …

Author Response

We greatly appreciate the reviewer´s comments, as the proposed suggestions will improve the understanding and the impact of this work. Accordingly, the revised version of the manuscript includes changes that are detailed below.

Major changes

- The reviewer suggested to include in the introduction elementary information about the biological consequences of the IL-23 downstream effector molecules. We agree with the reviewer that this information will greatly improve the understanding of the information discussed in this review. Thus, key information about IL-17, IL-22 and GM-CSF, their cognate receptors and signaling cascades, with a special emphasis on the biological functions and functional consequences of the action of these cytokines has been included in the introduction (lines 62-142).

- The reviewer suggested the incorporation of non-Immunoglobulin based strategies as alternative therapeutic intervention in IL-23 signaling cascade. The IL-23-Alphabodies (Desmet, 2014) were already included in the previous version, but this part has been completed with additional strategies involving protein scaffolds suggested by the reviewer (lines 182-191 and 292-304).

- The reviewer suggested to include a table summarizing all small molecules blockers discussed in the review. We agree with the reviewer that this type of table will greatly improve the impact of the review, providing a better overview of potential strategies. Accordingly, we have included this table in the revised version of the manuscript (new Table I).

- The reviewer suggested that the chapter regarding the interplay between IL-23 and AhR was too long in comparison to other parts of the review, and that key information about the biological activity of IL-22 was required to fully understand this part of the review. The major functional consequences of IL-22 action have been included in the introduction (lines 91-123). Regarding the length of the chapter, we believe that AhR function is relevant in the context of the IL-23 signaling cascade, but to a similar degree than other signaling molecules discussed in the review. Thus, this chapter has been reduced and included this part inside the Chapter 4: New players in IL-23 proximal signaling events (lines 633-672).

- The reviewer mentioned that IL-23 and IL-1β signaling crosstalk was too complex for a short review, and suggested to focus only on the most important recent findings or to completely remove this part. We believe this chapter is relevant in the context of IL-23 signaling cascade, as recent signaling adaptors such as mTOR or IRF4 have been identified specifically in the context of IL-23/IL-1b signaling, but we agree with the reviewer that this chapter was too long in comparison with the other chapters of the review. Accordingly, this chapter has been reduced in the revised version of the manuscript (lines 911-1348). We believe that IL-17 and IL-22 interfering strategies deserve an entire review on their own, and we feel that this topic has been already covered by recent and excellent reviews (Zwicky 2020, Dudakov 2015). These reviews are included in this review for further information for readers.

- Graphical representations. The reviewer mentioned that Figure 2 added low value to the review and accordingly, this figure has been removed. Regarding the suggestion of downloading particular structures from PDB database and including them in the review, in our experience, including figures with already published content that have not been specifically created for the manuscript is often problematic during technical revision of the manuscript. Thus, the structures have not been included, but the Protein Data Bank IDs for the crystal structures of IL-23 and IL-23/IL-23R and primary articles have been appropriately referenced (lines 205 and 275). As for the quality of the representations, we believe that graphical representation of signaling cascades are complicated, and we have tried to simplify graphics to provide a clear message. To offer a more dynamic view of the IL-23 signaling cascade we are currently preparing a 3D animated video abstract.

- Regarding the missing Greek lettering in the figures, it seems that there has been a problem during conversion to pdf format during submission, probably due to some kind of software incompatibility. Figures have been reinserted in a different format to fix the problem. Otherwise, these errors will be fixed during the technical revision of the manuscript.

- Minor typing errors have been corrected in the revised version of the manuscript.

Reviewer 3 Report

The manuscript by Pastor-Fernandez et al. summarizes the current knowledge about IL23 cytokine and its downstream signaling from the perspective of possible therapeutic interventions. The manuscript represents a comprehensive review and focuses on the highly clinically relevant signaling cascades. The manuscript is of potential interest, however additional modifications would be needed to further improve it and make it more balanced.

1). While the manuscript is entitled “Decoding IL23 signaling cascade for new therapeutic opportunities” some parts of this review (for example #7 IL23 and IL1b cross-talk) seems somewhat logically dissociated from the main focus of the review.  It would be beneficial for the coherence of the review to significantly reduced or take out this part.

2) Since the review is focused on the therapeutic potential, it would be essential to include more substantial discussion on other effects (pathogenic /side effects) of inhibition of IL23, adapter proteins or transcription factors, given their cell, tissue and disease-specific roles (STAT 3 and NFkb are the best examples with a variety of context-specific roles) . This will be very helpful for better representation of the complex IL23 biology.

3) Additional  references can be included

PMID: 16688182

PMID: 23034650

PMID: 30389414 

PMID: 29065479

PMID: 30721933

4) Minor: Symbol letters seems to become replaced on the figures  by boxes.

Author Response

We greatly appreciate the reviewer´s comments, as the proposed suggestions will improve the understanding and the impact of this work. Accordingly, the revised version of the manuscript includes changes that are detailed in a point-by-point response below.

Point-by-point response

- The reviewer suggested to reduce or take out the IL-23/IL-1b signaling crosstalk chapter, as it seemed dissociated from the review focus on IL-23 signaling cascade. We believe this chapter is relevant in the context of IL-23 signaling cascade, as recent signaling adaptors such as mTOR or IRF4 have been identified specifically in the context of IL-23/IL-1b signaling, but we agree with the reviewer that this chapter was too long in comparison with the other chapters of the review. Accordingly, this chapter has been reduced in the revised version of the manuscript (lines 911-1348).

- The reviewer suggested to include more discussion on potential side effects of some therapeutic strategies to interfere with IL-23 signaling cascade. We agree with the reviewer that this information will provide a better representation of the complexity of IL-23 signaling cascade and accordingly, the revised version includes potential risk of STAT3 and NF-kB inactivation in different tissues, with information obtained from patients with inherited defects in these signaling adaptors (lines 417-432 and 801-817).

- The reviewer suggested to include some references regarding the role of IL-23 in tumor models (PMID: 16688182, PMID: 23034650). The first publication was already included in the previous version of the review (concluding remarks, line 1368), and the second reference has now been added to this part (line 1369).

- The reviewer suggested the inclusion of some references discussing the associated comorbidities between psoriasis and cardiovascular diseases (PMID: 29065479, PMID: 30721933), and the role of IL-23 in atherosclerosis (PMID: 30389414). After exhaustive analysis of the available literature, we have not included a section discussing this, as the literature offers contradictory data, reflecting complex interplay between inflammation and cardiocascular events. PMID: 30389414 shows that genetic ablation of IL-23, IL-23R or IL-22 leads to deterioration of the intestinal barrier, expansion of pro-atherogenic bacteria and production of metabolites that promote macrophage activation and atherosclerosis in an atherosclerosis-prone Ldlr−/− mice model. Thus, this work suggests a protective role of IL-23/IL-22 axis in diet-induced atherosclerosis models. In contrast, PMID: 29065479 shows a large number of epidemiological studies demonstrating that psoriasis is associated with increased prevalence of cardiovascular diseases. This study analyzed how different anti-inflammatory treatments, including anti-IL-23 monoclonal antibodies therapies, impact on cardiovascular events with contradictory results. While some studies see protection (PMID: 30721933) or no impact (PMID: 25307931), other studies detected an increased risk of cardiovascular events upon anti-IL-23 treatment (PMID: 21862748, PMID: 22404103). We believe that the complex relationships between IL-23 and cardiovascular events deserves a specific topic review.

- Regarding the missing Greek lettering in the figures, it seems that there has been a problem during conversion to pdf format during submission, probably due to some kind of software incompatibility. Figures have been reinserted in a different format to fix the problem. Otherwise, these errors will be fixed during the technical revision of the manuscript.

Round 2

Reviewer 2 Report

I am fully satisfied with a substantial improvement presented in the revised version of the manuscript by authors. To my mind it is now ready for publishing.

Reviewer 3 Report

The authors performed extensive revisions of the manuscript, which significantly improved it. I have not further comments.